# ConnecSenS, a Versatile IoT Platform for Environment Monitoring: Bring Water to Cloud

**DOI:** 10.3390/s23062896

**Published:** 2023-03-07

**Authors:** Laure Moiroux-Arvis, Laurent Royer, David Sarramia, Gil De Sousa, Alexandre Claude, Delphine Latour, Erwan Roussel, Olivier Voldoire, Patrick Chardon, Richard Vandaële, Thierry Améglio, Jean-Pierre Chanet

**Affiliations:** 1Université Clermont Auvergne, INRAE, UR TSCF, 63178 Aubière, France; 2Laboratoire de Physique de Clermont, Université Clermont Auvergne, CNRS/IN2P3, 63000 Clermont-Ferrand, France; 3Université Clermont Auvergne, CNRS, Laboratoire Microorganismes: Genome, Environnement (LMGE), UMR, 6023 Clermont-Ferrand, France; 4CNRS, GEOLAB, Université Clermont Auvergne, 63145 Clermont-Ferrand, France; 5Université Clermont Auvergne, INRAE, PIAF, 63000 Clermont-Ferrand, France

**Keywords:** IoT, water, environmental monitoring, wireless sensor network, data integration, cloud, data stream, LoRaWAN

## Abstract

Climate change is having an increasingly rapid impact on ecosystems and particularly on the issue of water resources. The Internet of Things and communication technologies have now reached a level of maturity that allows sensors to be deployed more easily on sites to monitor them. The communicating node based on LoRaWAN technology presented in this article is open and allows the interfacing of numerous sensors for designing long-term environmental monitoring systems of isolated sites. The data integration in the cloud is ensured by a workflow driving the storage and indexing of data, allowing a simple and efficient use of the data for different users (scientists, administration, citizens) through specific dashboards and extractions. This article presents this infrastructure through environmental monitoring use cases related to water resources.

## 1. Introduction

Climate change and human activities are having an increasing impact on ecosystems. It is crucial to monitor them in order to understand the interactions between the many factors that disturb the balance. In this context, the issue of water is paramount: how to improve and maintain its quality [1]? How can we preserve the resource [2]? How can we optimize its uses [3]? The environmental monitoring system we developed in this work is specifically focused on water resource quality and management issues.

Sensor networks and, more generally, the Internet of Things (IoT) make it possible to envisage long-term monitoring of these environments. Indeed, the progress made in electronics over the last few years in terms of integration, energy sobriety, and cost reduction promotes the large-scale deployment of electronic devices. In addition, telecommunication technologies now allow the transmission of data over long distances (a few kilometers), even in the absence of heavy infrastructure and power sources.

Wireless sensor networks have revolutionized the field of environmental monitoring, allowing real-time data collection and analysis [4,5,6], but the solutions proposed are often theoretical or for short-term experiments, such as irrigation management [7]. Similarly, there is a lot of research in the literature on data storage in the cloud [8,9], but few works take into account the complete integration of the data from the sensor to the cloud over several years with various types of sensors and on several sites as we consider in our work. Nevertheless, there are works on the collection of long-term weather data as proposed by [10,11,12].

The latest research has focused on improving the accuracy and reliability of data collected through wireless sensor networks [13]. One of the major challenges in using wireless sensor networks is to ensure the power supply of sensors located in remote areas [14]. Our work is mainly focused on this challenge: How to have sensors working for a long time in outdoor conditions? 

Many experiments use commercial Arduino boards to make wireless nodes [15,16,17]. These devices have the advantage of being cheap, easy to implement, and used by a very large developer community. This technology is well suited for rapid and low-cost prototyping of embedded systems. However, the energy performance and footprint of these boards are not optimal for long-term outdoor experiments. [18,19]. Arduino-based platforms generally consist of a main microcontroller board with multiple extension modules. This design leads to a solution that is not always totally coherent at the level of size, energy management, and programming. It is why we have chosen to fully develop a specific node to optimize energy consumption in the context of use in constrained environments and be able to address big projects. This node also accepts a large variety of sensors to be able to instrument different measurement sites and to receive new sensors anticipating future applications related to environmental monitoring.

The main issue of our project is focused on the energy performance and longevity of the nodes. This issue is addressed in different ways in the literature. There are studies that focus on energy management at the node level through optimized operations of the node [20]. There are also approaches based on the optimization of information transfers, for example, by optimizing communication protocols [21,22], by compressing information [23], or by data aggregation [24]. Some solutions also propose embedding intelligence in the nodes where the knowledge is derived from a model [25] or learned via deep learning technologies [26]. In the literature, we also find some approaches to implementing energy harvesting systems [27] that increase the autonomy of the nodes at the cost of complexity and higher cost.

Within the framework of our project, we have designed a versatile, robust, and low-power node named Sensors open Lora node (SoLo).

In order to demonstrate the relevance of our solution, four sites were instrumented where water is a major issue: (1) a mountain lake subject to the pressure of agricultural practices; (2) a river channel connected to an oxbow lake whose hydroecological responses to climate change must be understood; (3) a former uranium mine close to a watercourse that can have an impact on the environment, and finally (4) a farm that uses water for its production. Figure 1 presents an overview of the system. These sites were chosen because they have been studied or monitored for many years [28,29,30], but research work is still in progress. The nature of the observations involves a wide variety of sensors (piezometer, temperature, soil and air humidity, radiation, etc.) which is a challenge to build a wireless sensor network with so many different sensors. These long-term studies allow us to evaluate the influence of climate change on these ecosystems: frequency of appearance of cyanobacteria, the evolution of water courses, temperature evolution, tree growth, etc.

## 2. Materials and Methods

### 2.1. Private LoraWAN Network

The topology of a private LoRaWAN (Long Range Wide Area Network) network deployed on the ConnecSenS sites is given in Figure 2. This technical solution assures low power consumption and a wide range of LoRa technologies. It is also an autonomous and independent solution, free of any support or charge with a network commercial operator. Each component of the network can be tuned according to the application requirements.

A star network is built with a gateway at the center, which receives the data frame from nodes scattered around the site. The communication between the nodes and the gateway is performed using LoRaWAN, a Low Power Wide Area Network (LPWAN) networking protocol based on the RF LoRa modulation technology operating, in this case, at 868 MHz. This technology was developed in 2014 by the French start-up company Cycleo, and today it is managed by Semtech [31]. It is based on a Chirp Spread Spectrum (CSS) modulation technique which encodes information using frequency chirps having a linear variation of frequency over time. This technology allows communication over long distances (several kilometers) [32] at low power (maximum effective isotropic radiated power of 40 mW in Europe) [33] but with a limited communication rate (50 kbps max.) [34]. The gateway transmits the data to a server via the Internet network, providing a bridge between the nodes (connected to the LoRaWAN network) and the server (connected to the Internet network).

### 2.2. Node SoLo (Sensors Open Lora Node)

#### 2.2.1. General Presentation of the Node

Due to the large variety of sites addressed by the present project and envisaged by future projects, as well as the number of different sensors deployed in the environmental sites, a generic LoRaWAN end node, named SoLo, has been developed. This node has been designed to allow data reading from a large variety of associated sensors, and it can be configured at the hardware and firmware levels to be adapted to the specific needs of measurements, autonomy, data processing, and data transmission. The hardware architecture of this node is given in Figure 3.

The SoLo node is built around an STM32 microcontroller [35], which controls all the operations thanks to embedded firmware developed in C and C++ languages. Each node has an internal SD memory to store sensor data, the history of the actions of the node (log file), and the configuration file. This configuration file, which is read by the microcontroller at start-up, is defined by the user and lists the configuration parameters of the node. A LoRa RF module is implemented on the electronic board to send data through the private LoRaWAN network. A GPS module has also been added to localize the sensor and to deliver a timing reference to the system. The power supply is delivered by a 3.6 V/9000 mAh battery cell managed by a charge controller circuit.

The SoLo node also integrates a small electronic card on which several sensors are implemented. These internal sensors can be activated or not. They measure temperature, air humidity, atmospheric pressure, luminosity, and acceleration. 

Several protocols of communication are implemented to communicate with external sensors: UART, SPI, SDI-12, I2C, analog and digital IO. These protocols correspond to a large variety of interfaces used by sensors. A slot implemented on the main electronic board allows the external sensors to be connected to the suitable input/output of the microcontroller.

The electronic boards and the battery cell are protected by an IP67 package/box (151 cm × 125 cm × 60 cm). On the external sides of the box, four M12 connectors allowing the interfacing of several sensors, a reset button, an SMA connector for the LoRa antenna, and a USB waterproof connector to interface a PC with the node are present. All these external connectors allow operations in the fields without needing to open the node (see Figure 4).

A main feature of the developed system is the possibility to easily configure its operations through a text file written on the SD card (“the configuration file” cited above). A list of the main parameters currently available is given in Table 1. Still, new ones can be added by modifying the firmware depending on the needs (new sensors, embedded data processing, new transmission strategy, etc.).

The energy autonomy of the node is a key issue in guaranteeing function for several months (ideally one year) without any human intervention. This autonomy depends mainly on the periodicity of the activation of the GPS module (mainly to synchronize the clock), the reading period of the sensors, and the delay between the LoRa transmissions. The SoLo node can thus be tuned to fit the supported application’s requirements accurately. It was validated on our sites: the nodes are still working in different places.

#### 2.2.2. Firmware Presentation

The block diagram of the architecture of the firmware implemented inside the node SoLo is given in Figure 5. Drivers allow reading the sensors whose data are both written in a CSV file on the SD card and in a buffer to temporarily store the data ready to send. Before their transmission, the data are encoded and encrypted inside a single LoRaWAN frame with a payload limited by the Data Rate (DR) value selected (51 bytes with DR = 0 up to 222 Bytes with DR = 7).

When the node initiates a LoRaWAN communication, an acknowledgment is returned by the server if it accepts the data to be sent. If the node does not receive this acknowledgment, it does not try again but stops communication until the next scheduled transmission time. Then, the previous data will be added to the new one within the limit of the payload. This mode of transmission saves energy when the conditions for the LoRa communication are not optimal. Still, it requires space in the payload to accumulate data in a single LoRaWAN frame.

### 2.3. Data Workflow

The data workflow infrastructure is composed of two main software modules: (1) the Network Management Module (NMM) and (2) the Data Management Module (DMM). The NMM focuses on LoRaWAN communication and architecture. LoRaWAN is one of the LPWAN technologies widely studied currently [31,36,37] with a MAC (Medium Access Control) layer protocol based on the LoRa (Long Range) physical layer. The NMM is built on a LoRaWAN stack such as ChirpStack [38] or The Things Stack [39]. According to this stack, a LoRaWAN gateway is equipped with software called the “packet forwarder.” The role of this software is to automatically forward all the received messages to the server. In basic configuration mode, the packet forwarder of a gateway accepts all the LoRaWAN frames, even those which are not related to its private network. The filtering/selection of the frames is realized in the higher-level software components of the LoRaWAN stack. In the case of the ChirpStack stack, these components are the “Gateway Bridge”, the “Network Server”, and the “Application Server”. These tools are responsible for selecting only the frames received from the accepted nodes. At the level of the “Application Server,” different Web interfaces are available, allowing, for example, the monitoring of the communication infrastructure (ex: the availability of the gateways) and the registration of authorized nodes. A single gateway can cover a wide area thanks to the performance of the LoRa radio modulation, but in large or very hilly fields, it is possible to deploy several gateways in a monitoring site to have better communication coverage. All these gateways can be associated with the same network server. In this network topology, no modification is required at the level of a gateway when a node is added or removed. This flexibility is a major asset for agricultural or environmental applications where changes to the network devices can frequently happen. Thus, nodes with different sensors can be added or removed depending on the required monitored parameters. Outdoor nodes are also submitted to extreme conditions, which increase the risks of failures and require “fast” node replacements.

Concerning the data, this NMM converts LoRaWAN frames of SoLo nodes into files generally in JSON (JavaScript Object Notation) format. The generated data files are then processed by the DMM of the data center infrastructure. This DMM has multiple functionalities, and it can be called “hybrid” because it combines tools from different software families. For example, it integrates database management systems (DBMS) in the NoSQL model (storing JSON and using corresponding SQL (Structured Query Language) queries) and more recent search engine technologies. As a DBMS, we can mention PostgreSQL [17] with PostGIS extension and, as a search engine, Elasticsearch [18].

Other functionalities are related to data visualization with the availability of tools such as Grafana [40] and Kibana [41]. Data can also be processed as (input and output) streams using, for example, MQTT (Message Queuing Telemetry Transport) protocol [42]. Data can also be exported in new JSON format, CSV, and others. Depending on the data connectors and converters used, the possibilities are huge.

In order to achieve real-time data visualization and to address the most various cases of possible use, a near-real-time multi-pipeline architecture has been designed in a generic way. This architecture is then able to process data from various sensor deployments. Once the data JSON file is created or updated with a new measure, a lightweight shipper called Filebeat [43] forwards data to a data collector called Logstash [43]. This tool acts as a data streaming pipeline, that is, an Input-Filter-Output process that can ingest a multitude of data sources (Input), clean and enrich each event with some relevant information (Filter), and route data into a data lake (See Figure 6). Filebeat is a robust tool with stopping and resuming data transmission capabilities in case of a network outage. It can also slow down the transmission if there is an ingestion issue with Logstash. For its part, Logstash has data resiliency ability using persistent queues. This mechanism protects against data loss by storing each line of received data in an internal queue on disk. Altogether, these tools enable reliable and near real-time data management. Interestingly, it provides a data visualization allowing, for example, to immediately check the installation of a sensor or a node in situ.

The routing of each measurement to its right index location is done using an index naming convention in which the experiment name (present in the node file configuration as mentioned in Table 1), the node name, and the measurement date is used. This way, variables can be easily isolated and queried to generate time series. Although the ingested data format is JSON in the context of a SoLo node, the pipeline can also handle structured sources (such as databases) or semi-structured sources (such as flat CSV, text, and JSON files). More information about data integration can be found in our previous article [44].

In order to analyze the performance of the networks, we studied the data workflow, the battery consumption, the rate of data loss during data transmission and the received signal strength over a given period. These metrics are those classically used in the literature to qualify wireless sensor networks [45,46].

### 2.4. Monitoring Sites

Four experimental sites are considered in this work. They are located in the Auvergne-Rhone-Alpes French region. All the scientific topics addressed through the instrumentation of these sites are related to water issues: The Aydat site is a mountain lake subject to the impacts of agricultural practices and recurrent cyanobacterial proliferation;The Allier River site is a river channel connected to an oxbow lake whose hydroecological responses to climate change must be understood;The Roffin site is a former uranium mine that can have a long-term impact on a watercourse and its vegetation;The Montoldre site is a farm where water is an input to be optimized.

The choice of the location of the sensors on the sites is obviously a function of the phenomena that we wish to observe. Still, this choice is also informed by the good knowledge of the hydrological functioning of the sites resulting from the year of observation. For the lake, we observe the lake alimentation and the parameters of the water column in the lake to understand the evolution of the cyanobacteria population. For the river, we want to follow the hydrological and flood evolution and the impact on the biodiversity, more particularly on the development of trees, so we use piezometers and dendrometers. For the Roffin site, the idea is to quantify the potential impact of uranium rejects on water and vegetation. Finally, for the last site, we use soil moisture sensors and weather stations to monitor crop development.

#### 2.4.1. Aydat Lake

Site description

The Aydat lake is located in the French Massif Central (45°39′48.35′′ N; 2°59′11.79′′ E) at 837 m above sea level. This natural lake was created by the damming of the Veyre river by a basaltic lava flow of 8551 ± 400 cal yr BP ago. It is a small dimictic lake with a total area of 60 hectares, a maximum depth of 15 m, and a small catchment area of 16,800 hectares. It is a eutrophic lake with recurrent cyanobacterial proliferation [47], especially of *Dolichospermum macrosporum*. This lake belongs to the OLA network (Observatory of LAkes), which largely contributes to research in lacustrine ecosystems in France.

Scientific objectives

Since the 20th century, anthropogenic activities have constantly increased, leading to a deterioration of water quality resources. The accumulation of nutrients, mostly coming from anthropic activities, led to a massive cyanobacteria bloom, which impoverished the ecosystem. The proliferation of cyanobacteria has a direct impact on ecosystem functioning and is also dangerous for human and animal health due to their capacity to produce cyanotoxins [48]. The latter frequently leads to restrictions on water activities or fishing activities. These last years, the public bathing zone of the Aydat lake was closed for several weeks due to toxic cyanobacteria proliferation, which has negatively impacted the local economy. 

In order to reduce the cyanobacterial proliferation and restore the ecological functions of the lake, a wetland was created in 2012 [49]. As this natural solution needs time to be efficient, cyanobacteria are still present, and research is still necessary to better understand the dynamics of cyanobacteria and their potential cyanotoxins production. More globally, the objective of this research is to get an overall understanding of the lake’s hydroecological processes in order to better forecast its responses to current and future changes in natural conditions and human-induced pressures (including land use changes in its watershed and alteration of the water temperature conditions due to climate change).

Sensors

Three types of sensors are deployed on this site (see Figure 7):1–3: Aquatroll 200 data logger. This water level and water temperature probe is installed in the Veyre river, upstream and downstream of the lake. The water level measurement is based on a piezoresistive sensor, whereas the water temperature and the specific conductivity, the salinity, and the Total Dissolved Solids (TDS) are monitored using a balanced 4-electrode cell. The water discharge series can be computed using a rating curve calibrated with a gauging protocol.4: Hydrolab HL7 multiparameter sonde. The multiparameter sonde comprises eight sensors, including an electrical conductivity sensor, a Hach LDO^®^ Dissolved Oxygen Sensor, a temperature sensor, a turbidity sensor, chlorophyll-a sensor, a blue and green algae sensor, rhodamine sensor, and finally a pressure sensor for water depth measurement. The Hach LDO^®^ Dissolved Oxygen Sensor provides a measure with Luminescent Dissolved Oxygen (LDO) technology. The Hydrolab conductivity sensor uses four graphite electrodes in an open-cell design to provide highly accurate and reliable data. The sensor measures specific conductance, salinity, total dissolved solids (TDS), and resistivity. The conductivity sensor uses four graphite electrodes designed to be compliant with the ISO 7027 Turbidity Measurement Standard. The Hydrolab temperature sensor is a variable resistance thermistor (316 stainless steel for corrosion resistance). Hydrolab sondes are available with integrated pressure sensors that provide depth measurements. Data acquisition of each parameter takes place every hour.5: Temperature data logger. The HOBO data loggers record temperature with the high-frequency acquisition (i.e., 5 min), located in the middle point of the Aydat lake, every 20 cm from the water surface to 3 m deep. The upstream river temperature (Veyre) is also monitored with eight HOBO data loggers regularly distributed (every 1 km) from the headwater to the river mouth.

#### 2.4.2. Allier River

Site description

The Allier River (France) is one of the last remaining European unregulated rivers with highly dynamic meandering sections. It flows 421 km north from its source located at 1485 m a.s.l. in the south part of the Massif Central (Lozère, France) to its confluence with the Loire River at the Bec d’Allier. The hydrological and flood regime of the Allier River is considered close to natural or unregulated even though two dams (Poutès and Naussac) located in the upper basin partially affect the discharge [50]. Therefore, the overall Allier River represents an opportunity to investigate alluvial and riparian vegetation processes adjusting to climate and catchment changes. More precisely, the study site is located around the Auzon Oxbow, one of the fluvial annexes of the Allier River in the upper river basin [51,52]. Oxbows are specific wetlands in the vicinity of river systems that play a crucial socio-economic and environmental role. Their ecosystem services are functionally efficient in regard to hydrological and ecological concerns: flood prevention and low flow mitigation, retention of excess nutrients, and refuge habitats for flora and fauna. Moreover, long-term monitoring of hydraulic annexes may then be considered as a relevant and synthetic proxy of the evolution of a wider river section (water and habitat quality, sediment connectivity, environmental resources, and changes in land use).

Scientific objectives

Recent climate change has caused significant changes in the Allier River hydrology. More specifically, an analysis of discharge records shows an increase in both severity and duration of low flow during the summer period and a decline by 10% of higher peak flows [50,53,54,55,56]. In addition, a long history of gravel mining activity within the Allier River led to a sedimentary deficit and a mean channel incision of the riverbed ranging between 1 and 1.5 m. Consequently, the altimetric level of the alluvial water table connected to the river channel may also have lowered, threatening the water availability for the riparian forest over the floodplain.

In this specific context, we designed a suitable monitoring system for the Auzon Oxbow site (see Figure 8) in order to evaluate the effects of droughts and severe low flow on the riparian forest growth, with a special focus on three plots of black poplars populations with contrasted sources of water uptake (groundwater, oxbow lake, river channel). 

Sensors

The ConnecSens monitoring system of the Auzon Oxbow constantly records three types of environmental measurements over the three studied plots of black poplar populations. (1) The water level and water temperature of each water source (water table, oxbow lake, river channel) with an Aquatroll 200 data logger; (2) the growth of three black poplar individuals per surveyed population using the PepiPiaf microdendrometric sensors [57]; (3) microclimatic data for each black poplar plots as air temperature, humidity and radiation through the internal sensors of the node SoLo.

#### 2.4.3. Roffin Mine

Site description

The former uranium mine of Roffin is located on the Gourgeat watershed in Lachaux (Puy de Dôme, France). Underground mining began in 1946, and in 1948 a treatment plant was built where the first French industrial processes for extracting uranium from ore were tested. Several settling ponds were also created to collect the residues from the ore treatment. The absence of significant veins, the difficulties of extraction, and above all, the discovery in 1952 of the important Bois Noirs deposit, located about ten kilometers away, led to the abandonment of mining in 1956 and the closing down and dismantling of the plant in 1958. The concession was given up in 1976. All in all, this first French uranium mine operated for about ten years and produced a mere 30 tons of uranium.

The more or less radioactive mine waste rock was left in place, and the highly radioactive ore processing residues (30,000 T) were abandoned in the old settling ponds. Over the years, vegetation has taken over the area without any significant human intervention since then, which is a unique case in France. Thus, the Roffin site is an in-situ laboratory (ZATU) for studying the medium- and long-term consequences of former uranium mines on ecosystems and the effects of low doses of radioactivity on living beings [58].

Scientific objectives

The objective of the instrumentation is to collect sufficient data to understand the transport and transfer of radionuclides in the different environmental compartments.

The two associated vectors are air and water. For air, the parameter monitored is radon in order to determine the exposure of the ecosystem to radon gas and to understand the exhalation fluxes from the ground. For water, the aim is to monitor the entire watershed to obtain a hydrogeological model (rainfall, surface, and groundwater flows) and understand the exchanges between surface and groundwater. This model will be a major input for radionuclide transport simulations. The collection of a very large number of long-term data will allow us to obtain a reliable hydrogeological model and validate the results of the radionuclide transport simulation tools.

Sensors

The monitoring system is based on four kinds of sensors: (a) the Aquatroll 200 for electrical conductivity, temperature, pressure, and water level; (b) rain gauges, (c) weather station; and (d) for radon gas, AERTT+©.

The deployment of the sensor network on the site faces several communication difficulties: the topography (canyon), the vegetation, and the distance from the GSM network antennas that involves a distance between the wetland and the gateway of more than 300 m (see Figure 9).

#### 2.4.4. Montoldre Farm

Site description

The INRAE site of Montoldre is one of the real conditions deployment sites of the network and data infrastructure presented in the previous sections. The Montoldre farm is located in the department of Allier in the Auvergne-Rhône-Alpes region at 300 m above sea level. The main goal of this monitoring site is to study the water in an agricultural use case. The network is based on different SoLo nodes connected to a local gateway through the LoRaWAN communication protocol. One of the particularities of this deployment is using an Ethernet connection between the gateway and the LoRa Server instead of a mobile technology protocol found in other sites.

Scientific objectives

In the Montoldre monitoring site, an experiment was conducted for months with the acquisition of data related to air and soil elements. Water is studied as the essential element for the cultivation of agricultural plants. Sensors are also used in the context of this experimental farm to have information on soil conditions for machines or robots working in the fields. The performance of node SoLo is also tested on this site: robustness of the communication, autonomy, etc. (see Figure 10).

Air temperature, humidity, and luminosity measurements are collected using the embedded sensors of the SoLo nodes. For the soil temperature and moisture, SoLo nodes are equipped with Truebner SMT-100 sensors [59]; see Figure 11. With these low-cost soil sensors, the goal is to get soil dynamics more than accurate measurements.

Node operating indicators are also monitored, such as battery level, RSSI (Receive Signal Strength Indicator), and SNR (Signal-to-Noise Ratio) for quality of transmission. For this experiment, SoLo nodes are located at different distances to test the impact of a LoRaWAN star topology network with heterogeneous ranges. Indeed, some nodes are localized at 50 m of the Montoldre LoRaWAN gateway, which is very close if we consider the LoRa capabilities. Other nodes are placed at 500 m of the gateway. The goal is not to reach a maximum transmission range but to study the possible impacts of spreading the nodes through different conditions at the level of an experimental farm.

## 3. Results and Discussion

### 3.1. Real-Time Visualization of the Data

For data visualization, a dedicated interface was built for each data-collecting project with a possible mix of dashboards, mostly graphs and data tables. A collecting project can go from a group of SoLo nodes in the same or in different areas of a specific monitoring site. The following Figure 12 shows an example of the data visualization interface.

For example, this interface allows comparing measurements from different SoLo nodes equipped with the same sensors. Comparison between different areas or experimental deployment conditions can be made. Here, the graphs and tables are complementary. Graphs provide a “rapid look” at the measurements and their evolution over time. Tables allow focusing on a given date or period with visualization of the data values. Data visualization is done by time series generated by queries on quantities (variables). Strikingly, the simplicity of the queries allows all users to create their own graphs. Secure access through the use of a personal account (a local account on the platform or the eduGAIN personal account by operating OpenSAML authentication) and the https (HyperText Transfer Protocol Secure) protocol enables data isolation, i.e., providing everyone with access only to the data they need and they are allowed to access. This is achieved with the combined use of group and organization with data sources in Grafana. An interesting point is potentially creating data sources for particular events by reindexing existing data reduced to defined variables, time intervals, or material. This reindexed source can then be deleted once the event is completed. Finally, although focusing on the real-time aspect, the dashboards can also be used for more sophisticated queries, such as cross-referencing or historical data. In this case, these graphs retrieved data from sources built upon PostgreSQL databases.

### 3.2. Data Analysis

#### 3.2.1. Battery Lifetime Estimation

SoLo node battery voltage monitoring, through the LoRaWAN communication network, allows the detection of power failures and then triggers on-site maintenance. The battery life can also be estimated to anticipate the maintenance operations, but the modeling of the battery discharge must integrate several variables, such as:The power consumption of the LoRa radio transceiver depends on the modulation parameters. Therefore, the channel frequency is set to 868 MHz, the bandwidth to 125 kHz, and the transmission power fixed to 25 mW. However, the spreading factor (7 to 12) can be selected through the data rate (5 to 0) parameter inside the configuration file;The power consumption of the internal sensors which the operator has activated;The power consumption of external sensors if the node itself provides the power;The period of data acquisition;The period of data transmission;The temperature of the battery cells has a significant impact on their discharge capacity (70% at 0 °C, 40% at −10 °C).

Results presented in [60] give some guidelines concerning the estimation and optimization of battery life for LoRaWAN communications, but they must be reconsidered in light of each use case. Another way to estimate the battery life is to measure the initial discharge slope of the battery.

Figure 13 shows the measurements of the discharge of a battery (lithium-ION 8800 mAh) for two different currents (A and B). The two curves (curves A and B) have the same shape with a fast decrease of the voltage during about one-third of the total discharge time of the battery, followed by a slower slope. The end of the battery life is briefly preceded by a very fast voltage drop. We can also observe that the steepness of the initial slope depends on the battery’s lifetime.

Using a dedicated test bench, several discharge curves were obtained by connecting the battery to different values of resistors. For each, the initial slope and the battery lifetime have been measured. The results are plotted in Figure 14. Two pairs of values extracted from data transmitted by nodes deployed on-site have also been added.

Figure 14 shows a good correlation between the value of the initial slope of the discharge and the battery lifetime. Using the regression equation $y=2\times 10^{-5}\times x^{-0.944}$, the value of the battery lifetime (y) can be evaluated by measuring the initial slope of the discharge (x) thanks to the data transmitted by the LoRaWAN network. Even if the precision of this estimation is limited to about 20% and does not consider the impact of significant temperature variations during the deployment of the node on site, this method is simple, robust, and useful for scheduling the maintenance period.

#### 3.2.2. Packet Loss Analysis

In a wireless sensors network, the analysis of the Packet Loss Rate (PLR) is crucial to assess the reliability of the network. Several factors affect the PLR, such as the weakness of the radio signal, the radio interference, the transmission distance, and the radio screening introduced by vegetation, buildings, and topography. The LoRa technology, integrated into the ConnecSenS network, presents a feature that can help to increase the communication range. Depending on the value of the Data Rate (DR) parameter, the communication range between the nodes and the gateway can cover a more (DR = 0) or less (DR = 5) long distance. However, as shown in Table 2, the Data Rate affects the bit rate and the acceptable sensitivity level (measured as the Signal-to-Noise Ratio, SNR). Decreasing the DR value decreases the bit rate, increasing the packet transmission time. But at the same time, it improves the sensitivity, which increases the communication range.

An analysis of the data collected on the field allowed us to extract the Packet Retransmitted Rate (PRR) and the PLR for each node. The PRR is the percentage of packets not received at their first sending. The PLR is the number of packets lost, computed as the total number of packets theoretically transmitted minus the number of received parquets out of the total number of packets theoretically transmitted. The results are presented in Table 3.

Depending on the nodes, the packet retransmission rate changes from 4% to 54%. The values presented in Table 3 show the benefit of retransmission with a PLR decrease of 11.3% on average on the nodes. On the Montoldre site, a second measurement campaign was carried out by changing the DR value of nodes 6201 and 6212 from DR = 5 to DR = 4. As presented in Table 4, this has increased the communication range with significant improvement of the PLR from 9% to 0% and from 11% to 0% for nodes 6201 and 6212, respectively.

In Table 3, we can observe that on the Auzon site, node 6203 has a much higher PLR than the other two nodes deployed on the same site. However, the three nodes have the same type of sensor connected (Aquatroll 200) and the same measurement period (1 h), meaning the same payload value. The difference is that node 6203 has a sending period of 4 h instead of 2 h for nodes 6234 and 6237. For the second time, the sending period of node 6203 has been configured to transmit every 2 h. The results, presented in Table 5, show that by transmitting the data every 2 h instead of every 4 h, the PLR falls from 31% to 2%. These results show the limits related to the size of the payload imposed by the LoRaWAN standard (payload of 51 bytes max for DR = 0 and up to 222 bytes max for DR = 5). It shows that if no additional space is available in the LoRa frame for previously untransmitted data, they will be lost. The PLR is not improved despite the retransmission feature implemented inside node SoLo. Increasing the transmission frequencies in order to save energy can have, on the opposite, a negative impact in certain conditions by increasing the number of retransmissions.

This analysis shows that the transmit period and the DR parameter are interesting levers for reducing the packet loss rate. The optimal solution can be calculated theoretically if the size of the payload is known. If not, an experimental analysis can be carried out to find the best compromise.

#### 3.2.3. Analysis of LoRa Signal Strengths Attenuation as a Function of Distance and Visibility Parameters (Path-Loss Model)

The Received Signal Strength Indicator (RSSI) of the LoRaWAN can be seen as a function of the transmission frequency and antenna properties, but mainly the distance between transmitters and receivers. The most common path-loss model is the log-normal model, which is given by [61,62]:(1)$$D=K-10\gamma \;log_{10}\left(\frac{d}{d_{0}}\right)$$
(2)$$\mathrm{With}\;K=P_{t}-C$$
(3)$$\mathrm{and}\;C=20\;log_{10}\left(\frac{\lambda }{4\pi d_{0}}\right)$$
where $d_{0}$ is a reference distance from the transmitter. *γ* is the signal power loss coefficient (PLE). *K* is a constant governed by the operating frequency and power of the antenna: $P_{t}$ being the transmitted power, *λ* being the wavelength of the signal.

Moreover, the received signal depends on many components, with visibility (Line-of-Sight or LOS) being the strongest. As far as we know, three LOS parameters may impact the signal attenuation: (1) whether the nodes are actually within the visibility range of the gateways; (2) the topographic configuration of the LOS; (3) the extent of the LOS masked by the forest cover. Based on the data collected from the four study sites, the objective is to assess the respective statistical effect of the logarithm of the Euclidian distance between nodes and gateways (logD) and the three LOS parameters listed below (respectively VIEW, LOS_CONF, and LOS_F) over the signal strength (RSSI).

The RSSI timeseries were collected from the four study sites using 16 SoLo nodes from 3 July 2021 to 14 November 2021 for the Auzon site (N = 33,661 RSSI measurements distributed over four nodes), from 25 May 2021 to 20 December 2021 for the Aydat site (N = 33,405 distributed over five nodes), from 5 January 2021 to 20 April 2022 for the Montoldre site (N = 31,193 distributed over four nodes) and from 14 April 2021 to 23 May 2022 for the ZATU site (N = 2374 distributed over three nodes).

The LogD parameter is computed as the decimal logarithm of the Euclidian distance between each node and its associated gateway. Euclidean distances between nodes and gateways are derived from their geographic locations expressed in Lambert 93 cartographic projection and collected in the field using a Trimble R10 GPS/GNSS device with centimeter accuracy. The viewshed of the gateways of the four study sites was computed using Digital Elevation Model (DEM) with a ground resolution of 1 m (RGE dataset from the French Institut National de l’information Géographique et forestière—IGN) and the *Viewshed* toolkit provided by the ArcGIS Pro software (3d Analyst Tools). Each node of the study site is then classified as *IN* or *OUT* of the viewshed of its gateway and stored in the VIEW bimodal variable of the dataset. The *LOS_CONF* parameter is derived from the LOS topographic profiles of each node which are extracted from the same RGE DEM using the *Stack Profile* tool provided by the ArcGIS Pro software (3d Analyst Tools). Each node is then classified into one of the three topographic configuration classes (A, B, or C) based on the open or broken nature of the LOS and its topographic mask index, according to Figure 15 The LOS_F_rate is the rate of the LOS covered by trees (>3 m high), computed from the 2019 aerial photographs provided by the CRAIG and the IGN.

A linear regression model was fitted to the RSSI data as a function of the logarithm of the Euclidean distance (logD) between the nodes and the gateway in order to quantify the effect of distance on LoRa signal fading. The residuals (i.e., variations in RSSI not induced by changes in distances between nodes and gateways) are then integrated into the dataset, and a random forest regression model is fitted to identify the visibility variables (VIEW, LOS_CONF, LOS_F_rate) that contribute the most to explaining the deviations from the standard lognormal path-loss model. Statistical comparison of RSSI between groups was also performed using the nonparametric Kruskal–Wallis test implemented in the *ggstatsplot* R package [63].

#### 3.2.4. RSSI Variation with the Geographical Distance to the Gateways

As expected, the RSSI data collected for the four study sites show a logarithmic degradation in signal strength with increasing geographical distance (Figure 16) between nodes and gateways. The average RSSI varies from about −32 dBm (node 6247 at Aydat, d = 1 m) in the immediate vicinity of the gateway to about −120 dBm (node 1276 at ZATU) when the node is located 260 m far from the gateway. The results of the statistical linear regression between RSSI and the logarithm of the node-gateway distance (logD) are robust and highly significant (R² = 0.808, *p*-value < 2.2 × 10^−16^, residual standard error = 10.68 dBm). Indeed, the linear regression model predicts an average decrease of 19.15 dBm (std. error = 0.03, *p*-value < 2.2 × 10^−16^) for an increase of one logD unit.

However, due to the high intra- and inter-node variance of the RSSI, the range of the linear regression residuals is extremely large. Some nodes exhibit highly platykurtic or even bimodal distributions of residuals (Figure 17). The standard deviation of residuals ranges from 1.89 for node 6217 (Montoldre) to 7.46 for node 6237 (Auzon). Nevertheless, the homoscedasticity criterion for residuals seems to remain unviolated. Nodes within the gateways viewsheds exhibit overall positive residuals (signal strength better than expected by the statistic path-loss model), except for node 1294 (the closest node to the Auzon gateway). Conversely, nodes outside the gateway viewsheds show negative RSSI residuals, with the exception of node 6212, which is located in the vicinity of the edge of the Montoldre Gateway viewshed (Figure 17B). In the same way, heavily broken LOS by topographic masks (LOS_CONF = C) exhibits negative RSSI residuals. In contrast, open LOS (LOS_CONF = A) shows overall positive residuals (still except for node 1294 in Auzon). The intermediate level of LOS topographic masking (LOS_CONF = B) exhibits moderate RSSI residuals, just between the two other LOS classes (Figure 17C). Increasing rates of LOS covered by forest seem to be associated with changing from positive to negative RSSI residuals (Figure 17D).

A random forest regression (number of trees = 1000, number of variables tried at each split = 3) was performed to assess the statistical contribution of the visibility variables (VIEW, LOS_CONF, LOS_F) to the explanation of the distribution of the residuals. The random forest regression model poorly explains the distribution of RSSI residuals with a Mean Of Squared Residuals (MSE) of 57.87 dBm and only 49.23 of the variance explained. The variable importance measured by the random forest (Figure 18) points out the VIEW (nodes in or out of the gateway viewshed) and LOS_F_rate (rate of LOS covered by forest) as the main visibility variables that explain the distribution of RSSI residuals with 96.21 and 23.64% of increase MSE (percent increase in the mean square error of the Random Forests model when the data for that variable were randomly permuted).

A partial dependence plot on VIEW clearly exhibits the deleterious effect on the signal strength of a location outside the gateway viewshed (Figure 18B). The partial dependence plot on LOS_F_rate shows that negative RSSI residuals are associated with LOS covered by forest beyond 23% (Figure 18B). Surprisingly, according to the random forest regression model, the LOS_CONF variable provides no additional information to explain the distribution of RSSI residuals (Figure 18A,B), even though the Kruskal–Wallis test reveals a significant difference in RSSI residuals between the three LOS topographic configuration classes (chi-square = 37,970, df = 2, *p*-value < 2.2× 10^−16^, and see violin chart Figure 19).

## 4. Conclusions

In this work, we have proposed a multi-purpose communicating node to acquire and transmit environmental data for several sensors. The deployment on-site allowed us to validate the robustness and reliability of this solution in different water resource management use cases. This reliability has also been successfully tested on the Etna site for the monitoring of volcanic activity [64]. The deployed nodes are still operational.

The proposed integration workflow based on free solutions has shown its relevance. The users appreciated the easy design of the dashboards allowing them to follow the activity of the sites and the status of the nodes (notably their battery level).

The analysis of the different nodes on each site shows heterogeneity in the quality of service rendered in terms of energy consumption, transmission quality, etc. A major issue is predicting the operation of the network, which is particularly important for remote sites. We have shown that it is possible to have an estimate of the battery life after a few days of operation thanks to the interpretation of the first evolution points of consumption. On the other hand, it is more complicated to predict the transmission quality a priori. Still, we have shown that the LoRa signal strengths attenuation can be evaluated considering the distance and the visibility parameters. Future dedicated experiments will provide additional data to make this modeling more robust.

Future work will also focus on deployments to other sites and on the issues of data quality: how to identify measurement drift and outliers and generate alerts accordingly? Smarter devices will also minimize data transmission and improve the battery lifetime.

## 5. Patents

The firmware of the SoLo node is protected by the following IDDN certificate: IDDN1.FR2.0013.2000104.0005.S6.P7.20208.0009.1020010, 13 May 2020.

## Figures and Tables

**Figure 1 sensors-23-02896-f001:**
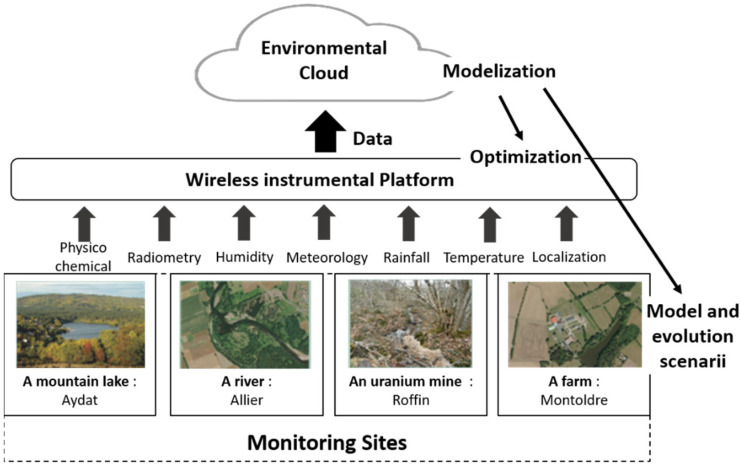
Synoptic of the network: Bring Water to Cloud.

**Figure 2 sensors-23-02896-f002:**
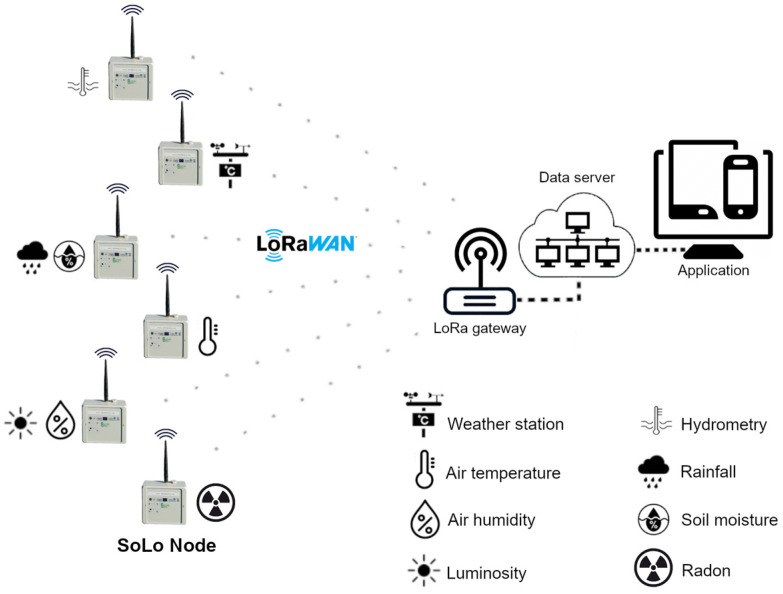
Private communication network based on a private LoRaWAN network to transmit data from the sensors to scientists.

**Figure 3 sensors-23-02896-f003:**
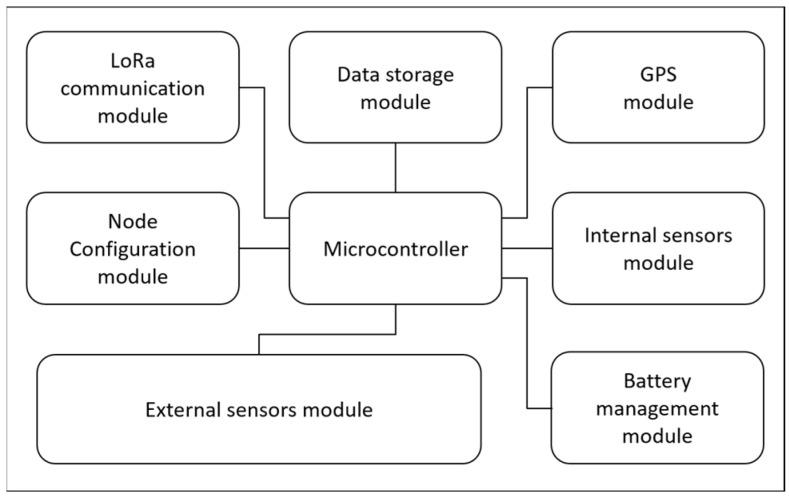
SoLo node hardware architecture built around an STM32 microcontroller.

**Figure 4 sensors-23-02896-f004:**
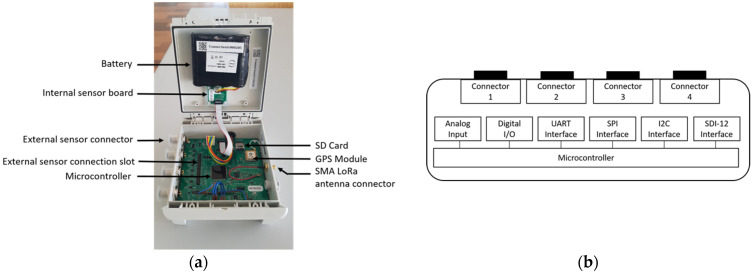
Electronic features of the SoLo node. (**a**) Electronic boards and batteries inside the box of the SoLo node. (**b**) Synoptic of the SoLo node I/O interface.

**Figure 5 sensors-23-02896-f005:**
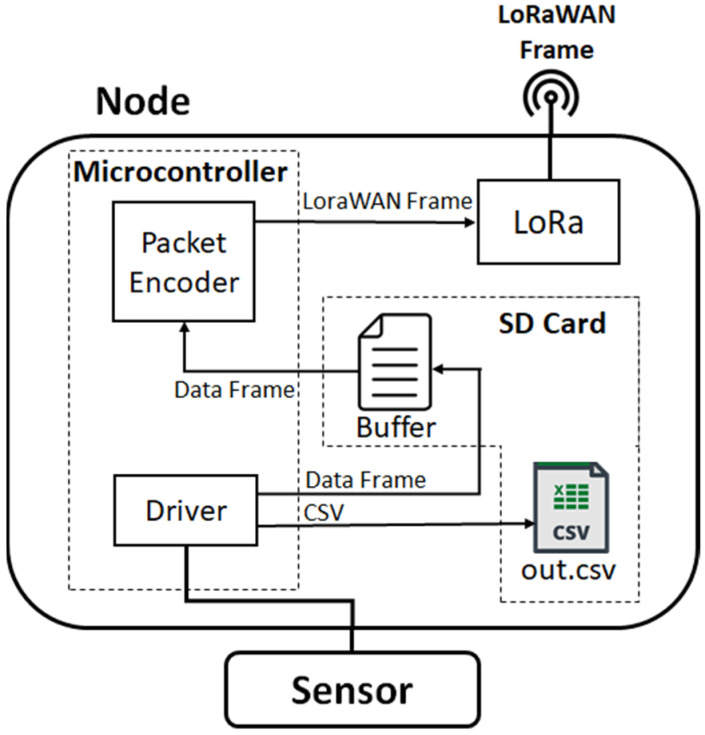
Architecture of the firmware implemented in the SoLo node: from sensor data reading to LoRaWAN transmission.

**Figure 6 sensors-23-02896-f006:**
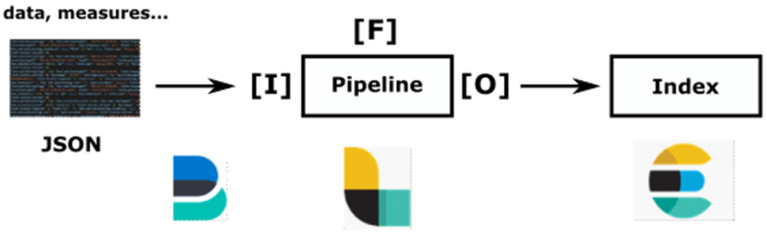
Input-Filter-Output process (from [44]).

**Figure 7 sensors-23-02896-f007:**
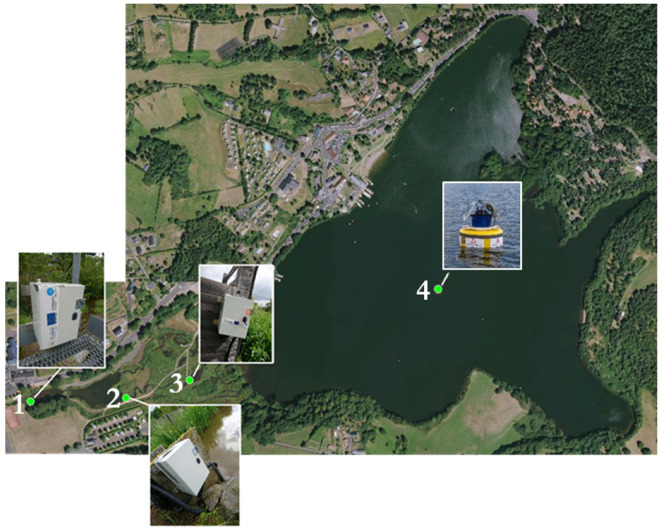
Location of sensors on Aydat site (1, 2, and 3: Aquatroll 200 data logger, 4: Hydrolab HL7) (Image ©2023 Google, Images ©2023 CNES/Airbus).

**Figure 8 sensors-23-02896-f008:**
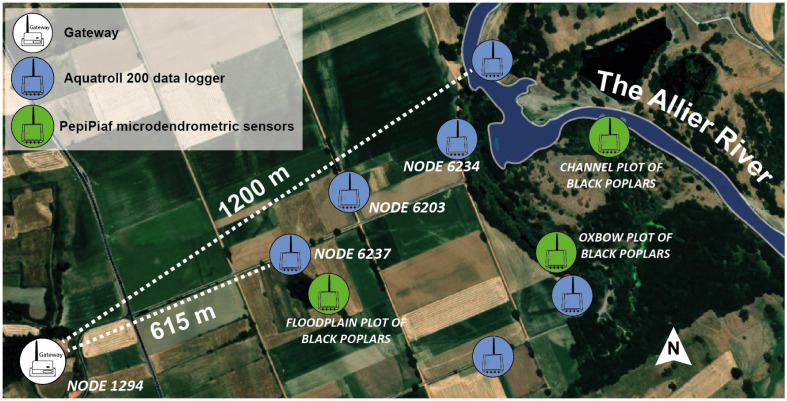
Location of sensors on the Auzon site (Image ©2023 Google, Images ©2023 CNES/Airbus).

**Figure 9 sensors-23-02896-f009:**
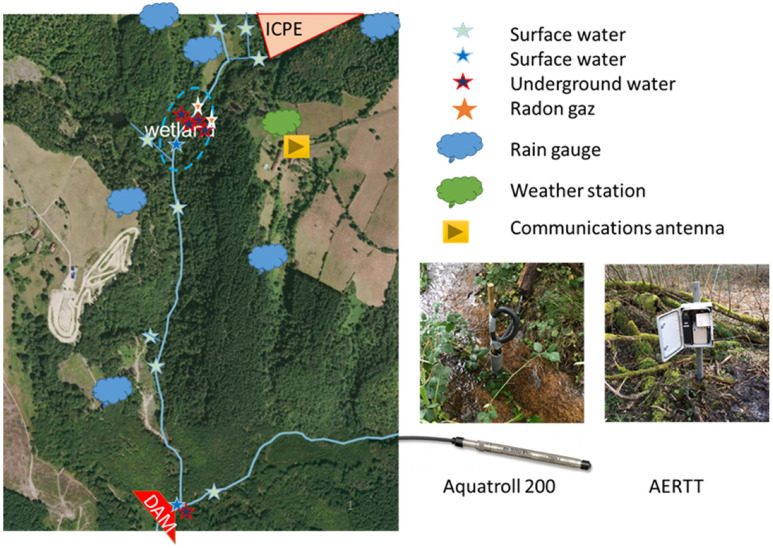
Overview of the Roffin site (Image ©2023 Google, Images ©2023 CNES/Airbus).

**Figure 10 sensors-23-02896-f010:**
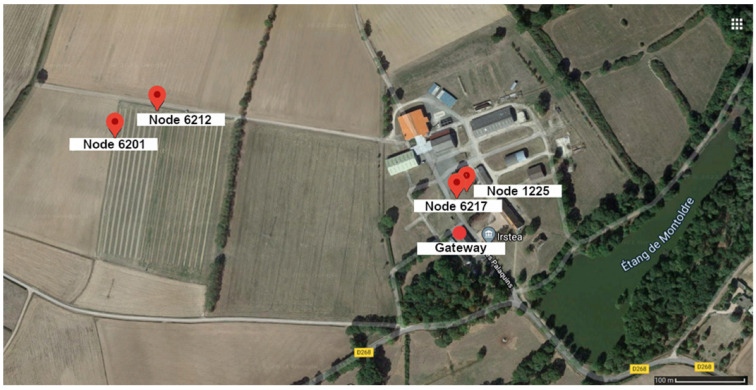
Location of nodes and the gateway on Montoldre site 3.4.3. Sensors (Image ©2023 Google, Images ©2023 CNES/Airbus).

**Figure 11 sensors-23-02896-f011:**
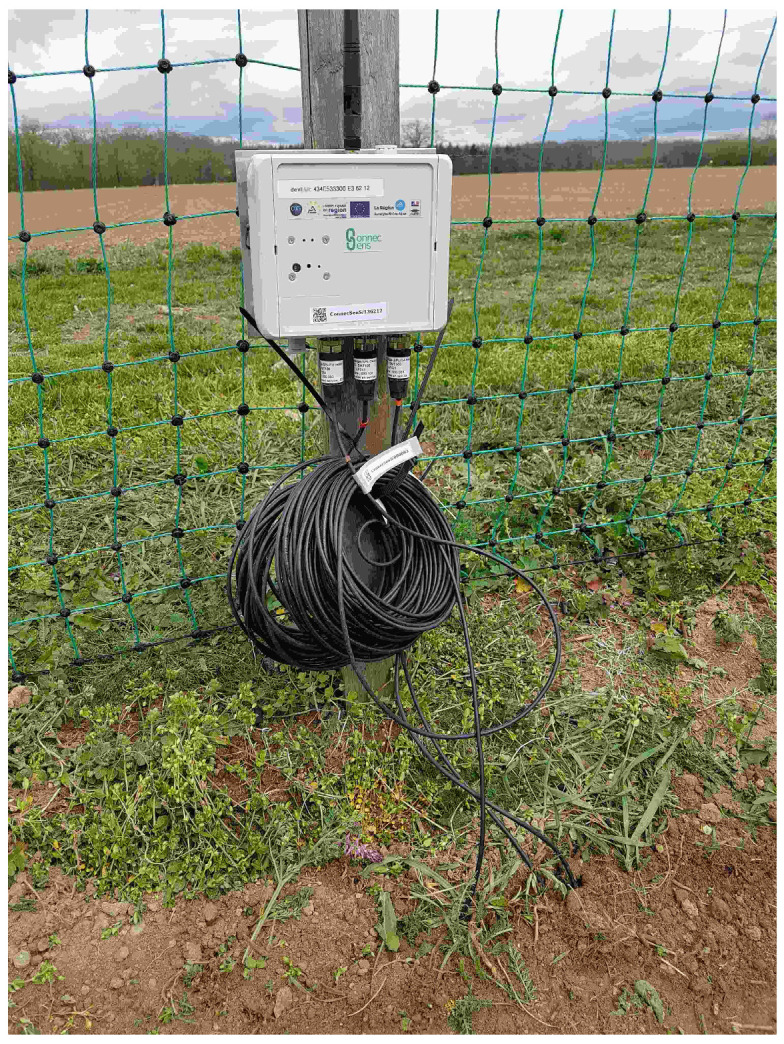
SoLo nodes deployed on the Montoldre site.

**Figure 12 sensors-23-02896-f012:**
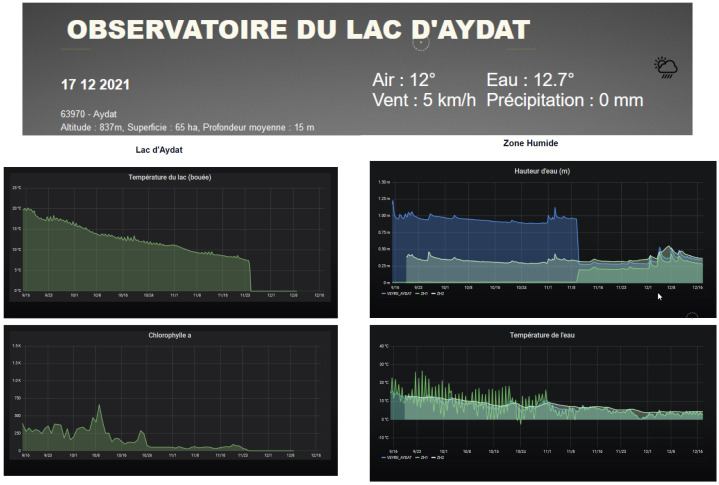
Example of data visualization interface for the Aydat site with an instantaneous KPI (Key Performance Indicator) board on the top and evolution graphs built with Grafana on the bottom.

**Figure 13 sensors-23-02896-f013:**
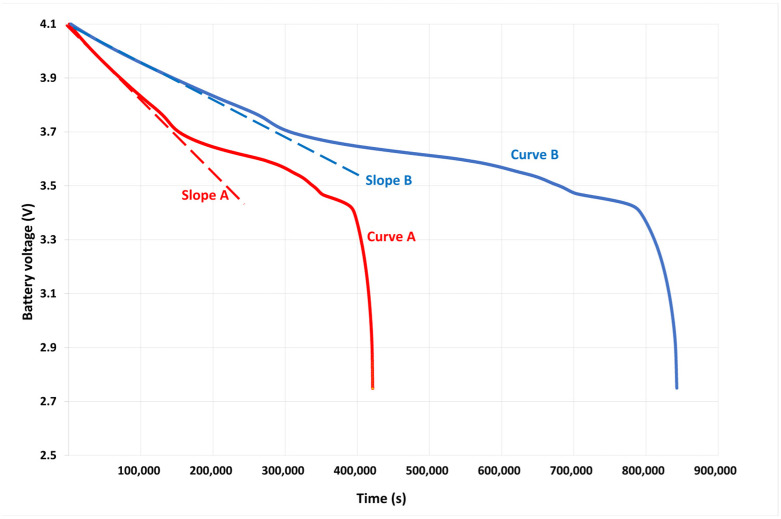
Evolution with the time of the battery voltage for two different currents (curves A and B) and initial discharge slopes (slopes A and B).

**Figure 14 sensors-23-02896-f014:**
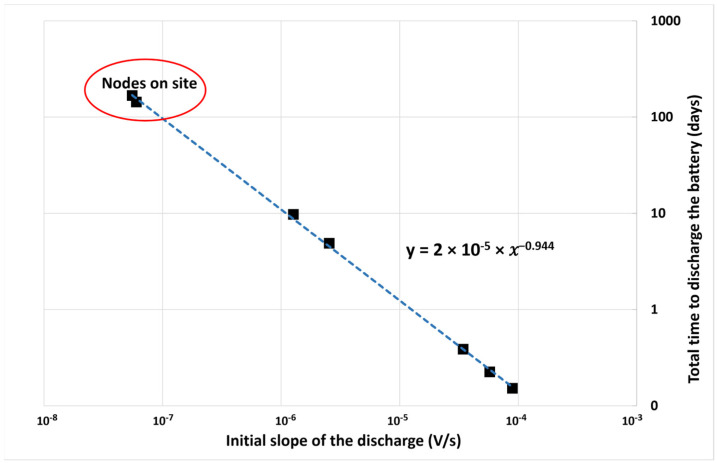
Correlation between the battery lifetime and the value of the initial slope of the discharge.

**Figure 15 sensors-23-02896-f015:**
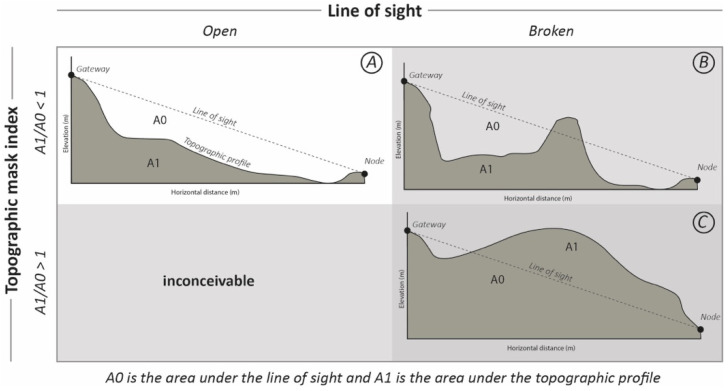
Classification classes for the nodes.

**Figure 16 sensors-23-02896-f016:**
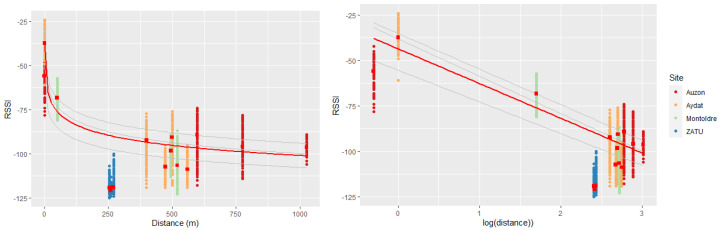
Statistical log-normal path-loss model (RSSI vs. Distance (m)).

**Figure 17 sensors-23-02896-f017:**
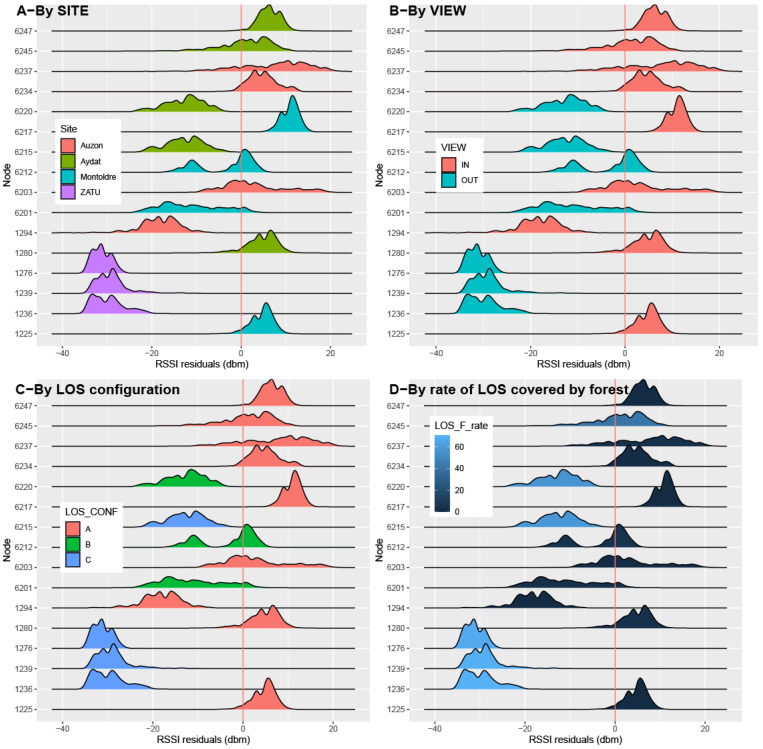
Regression residual analysis.

**Figure 18 sensors-23-02896-f018:**
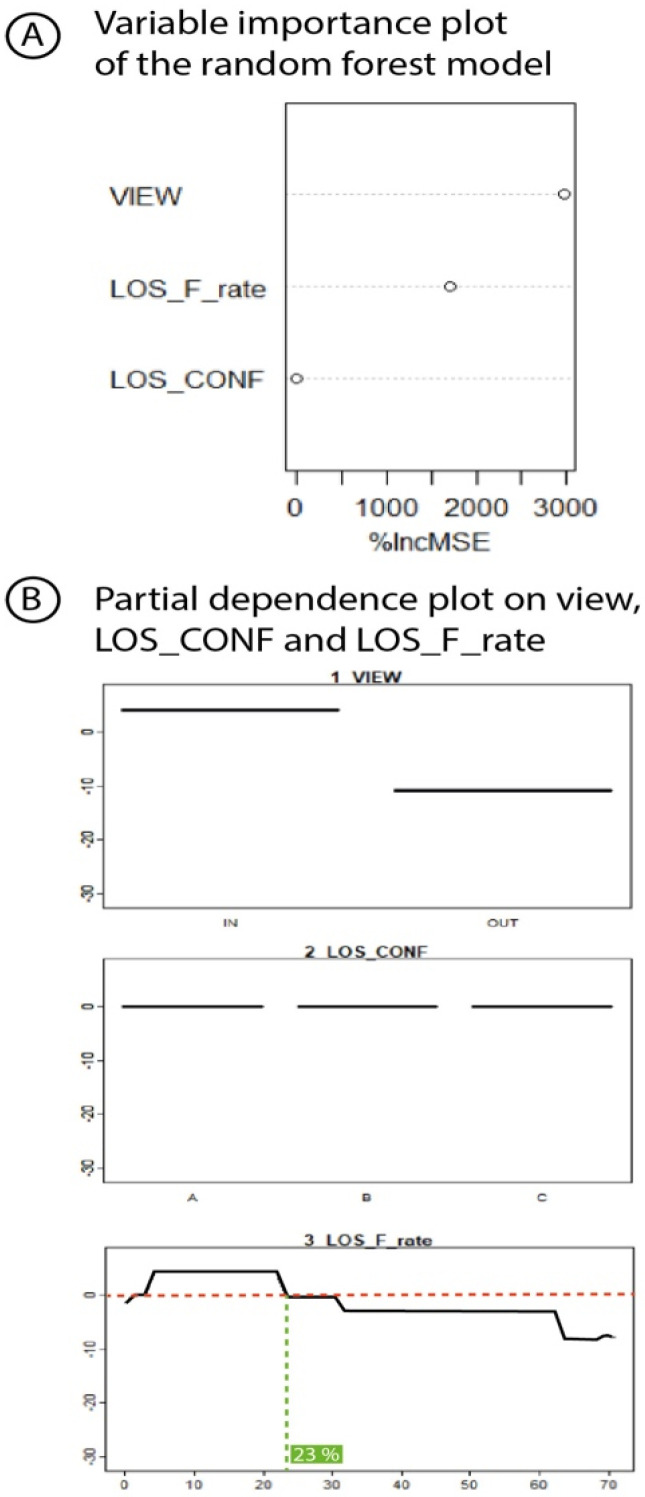
Random forest analysis.

**Figure 19 sensors-23-02896-f019:**
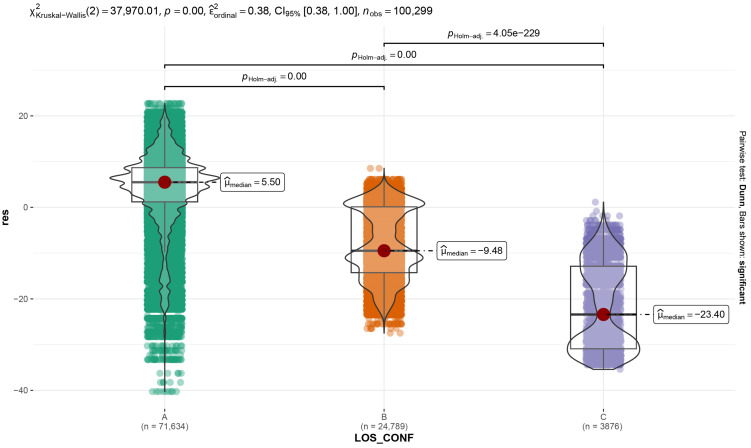
Kruskal–Wallis test result.

**Table 1 sensors-23-02896-t001:** Parameters to be defined in the configuration file.

General configuration	Experimentation name Node reference Log file information level Debug level, etc.
Sensors configuration	Declaration of the sensors interfaced to the node (internal and external sensor) Configuration of each sensor: type, measurement period, alarms, etc.
Network configuration	LoRaWAN parameters LoRa radio parameters settings (Data Rate) Transmission period
Time synchronization	GPS or manual

**Table 2 sensors-23-02896-t002:** Evolution of the bit rate and acceptable sensitivity depending on the data rate. BW = 125 kHz and CR = 4/5.

Data Rate (DR)	Bit Rate	Signal-to-Noise Ratio (SNR)
0	293 bit/s	−20 dB
1	537 bit/s	−17.5 dB
2	976 bit/s	−15 dB
3	1757 bit/s	−12 dB
4	3125 bit/s	−9 dB
5	5468 bit/s	−6 dB

**Table 3 sensors-23-02896-t003:** Packet retransmitted and packet loss rate for each node deployed on the experimental sites, as well as their configuration (connected sensors, transmit period, and data rate).

Site	Node	Connected Sensor to the Node	Transmit Period	Data Rate	Packet Retransmitted Rate	Packet Loss Rate
Montoldre	6201	3x SMT100	1 h	5	20%	9%
	6212	3x SMT100	1 h	5	24%	11%
	6217	Internal sensor 1x SMT100	1 h	5	14%	10%
	1225	Internal sensor	1 h	5	4%	0%
Auzon	6203	Aquatroll 200	4 h	5	35%	31%
	6234	Aquatroll 200	2 h	5	12%	0%
	6237	Aquatroll 200	2 h	5	14%	1%
Aydat	6215	Aquatroll 200	2 h	5	22%	9%
	6220	Aquatroll 200	1 h	5	5%	4%
ZATU	1236	Internal sensor Rain gauge	1 h	3	54%	35%
	1276	Aquatroll 200	1 h	3	28%	14%
	1239	Aquatroll 200	1 h	3	38%	10%

**Table 4 sensors-23-02896-t004:** Comparison of packet loss rate depending on DR setting.

Site	Node	Connected Sensor to the Node	Packet Loss Rate with DR = 5	Packet Loss Rate with DR = 4
Montoldre	6201	3x SMT100	9%	0%
	6212	3x SMT100	11%	0%

**Table 5 sensors-23-02896-t005:** Comparison of packet loss rate vs. transmit period setting.

Site	Node	Connected Sensor to the Node	Packet Loss Rate with Transmit Period = 4 h	Packet Loss Rate with Transmit Period = 2 h
Auzon	6203	Aquatroll 200	31%	2%

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
