# Peer review of "ConnecSenS, a Versatile IoT Platform for Environment Monitoring: Bring Water to Cloud"

_sensors, 2023, doi:10.3390/s23062896_

Round 1
Reviewer 1 Report
The manuscript has proposed a multi-purpose communicating node to acquire and transmit environmental data for a very large number of sensors. For the benefits of the reader, however, a number of points need clarifying and certain statements require future justification. There are given below.
1. Introduction: The authors do not provide detailed background information. There is not much introduction to the research progress of the existing literature, so the innovation of the manuscript and the research goals and significance are not clear.
2. The second paragraph of the introduction mentions that the progress made over the last few years has enabled the miniaturization of sensors and transmission of data over long distances. The fourth paragraph mentions that many experiments use Arduino modules as the core of the wireless nodes. However, because of its high energy consumption, it is not suitable for long-term outdoor experimentation. Because I don't know much about this, I feel a little contradictory.
3. Materials and Methods: Supplement with appropriate theory and literature as support.
4. Materials and Methods: Although the manuscript details the LoraWAN network, Node SoLo, and Data Workflow. But there are also some unimportant content that can easily cause confusion in the reader's thinking and logic.
5. Monitoring sites: What are the reasons why the manuscript chose four monitoring sites as study areas? What are they typical and representative? In particular, it is mentioned in the manuscript, climate change is having an increasingly rapid impact on ecosystems and particularly on the issue of water resources. What are the climatic characteristics of the four monitoring sites?
6. How does the manuscript determine where to deploy the various sensors? It is recommended that this information be supplemented to improve the clarity of the manuscript.
7. The manuscript mentions, there is a lot of work on the issue of monitoring the environment using wireless technologies, but often the solutions proposed are theoretical or for short-term experiments. However, the manuscript was only studied for 3 months (9.16 to 12.16).
8. There are typos in the manuscript. For example, nOde in line 95.
9. There are insufficient references for the last five years.
10. As required by Sensors, the manuscript should include introduction, materials and methods, result, discussion, and conclusion. Specific requirements can be found on the website: https://www.mdpi.com/journal/sensors/instructions#submission
Obviously, the authors also need to discuss and explain the findings, the impact of the results, and highlight the limitations of the work and future research directions. In addition, the manuscript has too many paragraphs, especially some of which are composed of one sentence (e.g., section 4.2.2).
Author Response
Introduction: The authors do not provide detailed background information. There is not much introduction to the research progress of the existing literature, so the innovation of the manuscript and the research goals and significance are not clear.
We have added three paragraphs to clarify the issues at stake in our research and the specific points on which our work is focused. We have supported our argument with recent bibliographic references as requested.
The second paragraph of the introduction mentions that the progress made years has enabled the miniaturization of sensors and transmission of data over long distances. The fourth paragraph mentions that many experiments use Arduino modules as the core of the wireless nodes. However, because of its high energy consumption, it is not suitable for long-term outdoor experimentation. Because I don’t know much about this, I feel a little contradictory.
We have added a sentence to precise that Arduino is not adapted to projet like ours. This finding is based on recent work and a general study (Kondaveeti, Hari Kishan, Nandeesh Kumar Kumaravelu, Sunny Dayal Vanambathina, Sudha Ellison Mathe, et Suseela Vappangi. « A Systematic Literature Review on Prototyping with Arduino: Applications, Challenges, Advantages, and Limitations ». Computer Science Review 40 (1 mai 2021): 100364.).
Materials and Methods: Supplement with appropriate theory and literature as support.
We have added elements to improve this section.
Materials and Methods: Although the manuscript details the LoraWAN network, Node SoLo, and Data Workflow. But there is also some unimportant content that can easily cause confusion in the reader’s thinking and logic.
We have removed some information not important for the paper.
Monitoring sites: What are the reasons why the manuscript chose four monitoring sites as study areas? What are they typical and representative? In particular, it is mentioned in the manuscript, climate change is having an increasingly rapid impact on ecosystems and particularly on the issue of water resources. What are the climatic characteristics of the four monitoring sites?
We have added the reason of the choice in a specific paragraph at the end of page 2.
How does the manuscript determine where to deploy the various sensors? It is recommended that this information be supplemented to improve the clarity of the manuscript.
A paragraph has been added at the beginning of section “Monitoring sites”
The manuscript mentions, there is a lot of work on the issue of monitoring the environment using wireless technologies, but often the solutions proposed are theoretical or for short-term experiments. However, the manuscript was only studied for 3 months (9.16 to 12.16)
Yes, the nodes are still working and we specified it at the end of the section 2.2.1.
There are typos in the manuscript. For example, nOde in line 95.
We made the corrections. Some, like SoLo, are voluntary as this is the official registered name of the node.
There are insufficient references for the last five years.
We have added more recent references in relation to your first remark.
As required by Sensors, the manuscript should include introduction, materials and methods, result, discussion, and conclusion. Specific requirements can be found on the website: https://www.mdpi.com/journal/sensors/instructions#submission.
We have changed the level of the section to follow the requirements.
Obviously, the authors also need to discuss and explain the findings, the impact of the results, and highlight the limitations of the work and future research directions.
We have improved the conclusion in that way.
In addition, the manuscript has too many paragraphs, especially some of which are composed of one sentence (e.g. section 4.2.2).
We have reduced the number of paragraphs as requested.

Reviewer 2 Report
I have added some comments and questions to the paper. Please find them.

Author Response
Line 21: Remove these points
Done
Line 67: I would suggest to remove this paragraph. I think it is not necessary.
Done
Line 81: I would suggest to add some peer-reviewed references for this paragraph.
We added several recent references to be more precise.
Line 147: Have you experimented this one year period in different conditions? or it is just your assumption?
Yes, we have experienced it: we write a sentence in that way.
Line 169: My knowledge in this subsection is very limited. I would suggest that other reviewers have a more careful look into this subsection.
This comment is not for the authors.

Round 2
Reviewer 1 Report
The revised manuscript is significantly improved. Authors have responded my concerns. I think it is acceptable for publication.
Reviewer 2 Report
I suggest that in future works, the data of this system be integrated with satellite data so that more robust spatial-temporal data can be created for the region.